# Local LoRA: Memory-Efficient Fine-Tuning of Large Language Models

**Oscar Key**[1*]     **Jean Kaddour**[1*]     **Pasquale Minervini**[2]
[1]Centre for Artificial Intelligence, University College London
[2]School of Informatics, University of Edinburgh
{oscar.key.20, jean.kaddour.20}@ucl.ac.uk
p.minervini@ed.ac.uk

## Abstract

We present Local LoRA, a memory-*flexible* fine-tuning approach that, in principle, can fine-tune an arbitrarily large model on fixed hardware, including consumer-grade GPUs. Our approach aims to decouple the size of the model and the memory required to fine-tune it by dividing the model into chunks and sequentially fine-tuning each chunk. Our results show that Local LoRA closes the gap between the un-tuned model and end-to-end LoRA on math reasoning tasks.

## 1 Introduction

Given the growing prevalence of open-source LLMs (HuggingFace, 2023), end users are increasingly interested in fine-tuning them using task-specific datasets. As the performance of LLMs improves as the model size gets larger (Kaplan et al., 2020; Wei et al., 2022), there is a strong incentive for the end user to download and fine-tune the largest models possible given available hardware.

The main bottleneck in this process is the memory consumed by fine-tuning the model, as end users may be unable to afford the memory resources required for this task (Schwartz et al., 2020). This has resulted in a large amount of work on reducing the memory footprint of the fine-tuning process, allowing it to be performed on consumer hardware. For example, lowering the numerical precision (Micikevicius et al., 2018; Kalamkar et al., 2019), check-pointing gradients (Chen et al., 2016), and parameter-efficient fine-tuning (PEFT) methods (Houlsby et al., 2019) aim to reduce these overheads.

Low-Rank Adapter (LoRA, Hu et al., 2021) fine-tuning is a parameter-efficient method that learns only a small set of trainable update parameters (called *adapters*) while fixing the full model parameters. LoRA reduces GPU memory overheads by storing gradients and optimizer states for only the trainable parameters instead of all the model parameters. However, LoRA is still bottle-necked by the size of the model: the entire model must be evaluated during the forward pass and the gradients are back-propagated through the pre-trained model weights to the adapters. Thus, the entire pre-trained network must be stored in memory.

In this work, we aim to alleviate this bottleneck by fine-tuning a large model chunk-by-chunk, where each chunk is sized to fit in the available GPU memory. Inspired by the local learning literature, we introduce *Local LoRA*, which allows each chunk to be trained in isolation by using a local loss function to provide the gradient signal. This allows the user to train arbitrarily large models, although we expect the downstream performance of the model to fall as the chunk size decreases. Our preliminary results show that Local LoRA outperforms the unfine-tuned base model on several math problem-solving tasks but does not perform as well as End-to-End (E2E) LoRA when fine-tuning a

---

*Equal contribution

Workshop on Advancing Neural Network Training at 37th Conference on Neural Information Processing Systems (WANT@NeurIPS 2023).

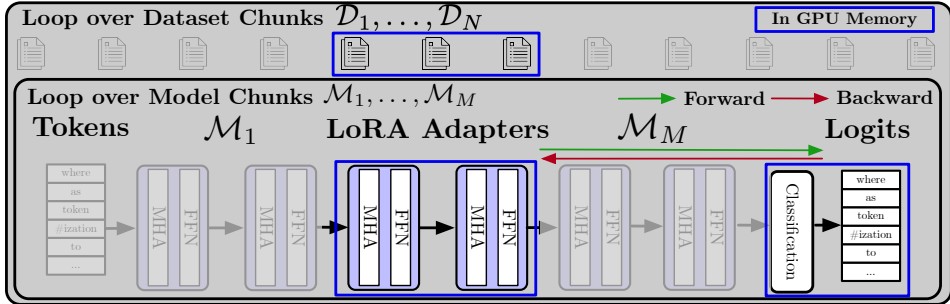

Figure 1: **Schematic Overview of Local LoRA.** In the outer loop, we loop over dataset chunks $\mathcal{D}_1, \ldots, \mathcal{D}_N$. For each chunk, we then sequentially update the model chunks $\mathcal{M}_1, \ldots, \mathcal{M}_M$. Here, the model chunks refer to LoRA adapters; in principle, they can be any sort of parameters to fine-tune. While fine-tuning a model chunk $\mathcal{M}_i$, we skip all forward/backward pass computations for $\{\mathcal{M}_j\}_{j>i}$.

model of the same size. However, when using Local LoRA to fine-tune a larger base model than is possible using E2E LoRA, the gap closes on all tasks and Local LoRA exceeds E2E on two tasks.

## 2 Background

In language model fine-tuning (FT), we update the pre-trained model $\boldsymbol{\theta}_{\text{PT}}$ to $\boldsymbol{\theta}_{\text{FT}}$ to maximize the language modeling objective given an FT dataset $\mathcal{D}_{\text{FT}}$, such that $\boldsymbol{\theta}_{\text{FT}} := \max_{\boldsymbol{\theta}} \sum_{\boldsymbol{x}^n \in \mathcal{D}_{\text{FT}}} \sum_{t=1}^{|\boldsymbol{x}^n|} \log \left( p_{\boldsymbol{\theta}} \left( x_t^n \mid x_{1:t-1}^n \right) \right)$, where $\boldsymbol{x}^n$ is a sequence of tokens.

One way to yield $\boldsymbol{\theta}_{\text{FT}}$ is vanilla fine-tuning of all layers. While simple and often the gold standard in terms of performance, this approach is memory-intensive. It requires storing the model, gradients of the target parameters, and, for the commonly used Adam(W) optimizer (Kingma & Ba, 2017; Loshchilov & Hutter, 2019), two moments of the gradients. In total, this is equivalent to storing four copies of the model in memory, which can become prohibitively expensive for large models.

A lightweight alternative is parameter-efficient fine-tuning (Houlsby et al., 2019), where we only fine-tune a small set of parameters. These methods are motivated by the observation that there often exists a low-dimensional reparameterization of pre-trained models that is as effective for fine-tuning as the entire parameter space (Aghajanyan et al., 2020). Because these methods only train a small set of parameters, the memory requirements, including optimizer states, are dramatically reduced.

### 2.1 LoRA

Low-Rank Adapter (LoRA) (Hu et al., 2021) fine-tunes low-rank-parameterized update matrices which are added to the fixed, pre-trained weights. For example, for a linear layer $\boldsymbol{Y} = \boldsymbol{X}\boldsymbol{W}$ with $\boldsymbol{X} \in \mathbb{R}^{b \times h}, \boldsymbol{W} \in \mathbb{R}^{h \times o}$, LoRA's modified forward pass yields $\boldsymbol{Y} = \boldsymbol{X}\boldsymbol{W} + s\boldsymbol{X}\boldsymbol{L}_1\boldsymbol{L}_2$, where $\boldsymbol{L}_1 \in \mathbb{R}^{h \times r}$ and $\boldsymbol{L}_2 \in \mathbb{R}^{r \times o}$ are the learned parameters, with $r \ll b$, and $s$ is a scalar hyper-parameter that can be tuned. To reduce the memory requirements of LoRA further, related work has explored the quantization of the pre-trained network (Dettmers et al., 2023) or adapters (Xu et al., 2023).

## 3 Local LoRA

We present a method for fine-tuning a large model chunk-by-chunk, enabling fine-tuning of a model larger than the available memory. While we focus on extending LoRA, it can also be combined with other PEFT and memory efficiency techniques (e.g. quantization and gradient checkpointing).

### 3.1 Model Chunking

Consider a Transformer model composed of $L$ blocks (Vaswani et al., 2017). We can view this as a composition of functions, $\mathcal{M} = d \circ f_L \circ f_{L-1} \circ \ldots \circ f_1 \circ e$, where $e$ is the embedding layer, $d$ is the de-embedding, and each $f_i$ is a Transformer block. Following ideas from the local learning

---

**Algorithm 1** Local LoRA

---

**Input:** model chunks $\{\mathcal{M}_i\}_{i=1}^C$, dataset chunks $\{\mathcal{D}_j\}_{j=1}^D$, n_epochs
**for** epoch $\in \{1, \ldots, \text{n\_epochs}\}$ **do**
    **for** $\mathcal{D}_j \in \{\mathcal{D}_j\}_{j=1}^D$ **do**
        **for** $\mathcal{M}_i \in \{\mathcal{M}_i\}_{i=1}^C$ **do**
            **if** first dataset chunk **then** load_pretrained_weights($\mathcal{M}_i$)
            **if** first model chunk **then** inputs $\leftarrow$ tokenize($\mathcal{D}_j$) **else** inputs $\leftarrow$ last_embeddings
            combined_model $\leftarrow$ get_proxy($\mathcal{M}_C$) $\circ\, \mathcal{M}_i$
            load_on_gpu(combined_model and optimizer states)
            last_embeddings $\leftarrow$ train(combined_model, $\mathcal{D}_j$)
            unload_from_gpu(combined_model and optimizer states)

---

and parallel training literature (Jaderberg et al., 2017; Mostafa et al., 2018; Nøkland & Eidnes, 2019; Belilovsky et al., 2020; Laskin et al., 2020), Local LoRA divides this model into chunks by grouping the functions like so,

$$\mathcal{M} = (d \circ f_L \circ \ldots \circ f_l) \circ (f_m \circ \ldots \circ f_n) \circ \ldots \circ (f_o \circ \ldots \circ f_1 \circ e),$$
$$\mathcal{M} = \mathcal{M}_C \circ \mathcal{M}_{C-1} \circ \ldots \circ \mathcal{M}_1,$$

where $l$,$m$,$n$, and $o$ are the layers at which the model is split, $C$ is the number of total chunks, and each $\mathcal{M}_i$ a chunk. By dividing the model into sufficiently many chunks, we can fit the parameters in the memory of consumer-grade GPUs with relatively small amounts of memory.

The idea is to fine-tune the model one chunk at a time, starting with $\mathcal{M}_1$, the chunk containing the embedding layer, and finishing with $\mathcal{M}_C$, the chunk containing the de-embedding. The algorithm loads each chunk onto the GPU in turn, performs several steps of fine-tuning, and then unloads it and loads the subsequent chunk. Figure 1 gives an overview and Algorithm 1 a detailed description.

**Dataset Chunking**    Naive local fine-tuning of model chunks is infeasible due to two problems.

First, for any chunk $\mathcal{M}_i$ with $i > 1$, we require the embeddings generated while fine-tuning $\mathcal{M}_{i-1}$ to use as the inputs to $\mathcal{M}_i$. These embeddings are large, potentially multiple TBs for an entire dataset, and thus difficult to store. Even if sufficient disk space is available, just writing the embeddings to disk can take a substantial amount of time.

Second, as we compute the embeddings while fine-tuning the previous chunk, the embeddings computed earlier will be "stale" as they have been computed using out-of-date parameters. Re-computing the embeddings after fine-tuning the chunk would be an overhead.

We address these issues by *dataset chunking*: we iterate through all the model chunks on the first dataset chunk, then switch to the second dataset chunk and iterate over all the model chunks again, and so on. This means we only have to store a fraction of the full set of embeddings at one time—in fact, we keep them in CPU memory—and the embeddings only become slightly stale.

**Proxy Model**    To compute the gradient of the output of chunk $\mathcal{M}_i$ we use a local loss function, implemented by a proxy model. The proxy model maps from the output of chunk $\mathcal{M}_i$ to the output of the full model, thus allowing the loss and gradient to be computed. The proxy should be small to conserve memory for the main chunk and reasonably approximate the gradient of the remainder of the model, ie, when training the $i$th chunk on input $h_{i-1}$ which outputs embeddings $h_i = \mathcal{M}_i(h_{i-1})$, the proxy model $\mathcal{P}_i$ should ideally satisfy $\partial \mathcal{P}_i(h_i)/\partial h_i \approx \partial(d \circ \mathcal{M}_C \circ \ldots \circ \mathcal{M}_{i+1})(h_i)/\partial h_i$.

Many choices of the proxy model are possible, but in this work, we make the simple choice of using the final hidden layer of the model and the de-embedding, $\mathcal{P}_i = d \circ f_L$. This choice is motivated by works on layer-dropping (Fan et al., 2020; Zhang & He, 2020; Kaddour et al., 2023b), which found that Transformer blocks can be dropped during training without destabilizing the model. Our proxy model is equivalent to dropping intermediate layers between the chunk and the output. Alternatively, it has been shown that learned linear projections on top of early hidden states can output embeddings very close to the final ones (Belrose et al., 2023).

Table 1: **Validation loss on MathInstruct and downstream evaluation on math problem solving tasks.** "Base": unfine-tuned pre-trained model. Val. loss: lower better. Other tasks: higher better.

| Model | | val. loss | Open-Ended | | | | | | Multiple-Choice | | |
|---|---|---|---|---|---|---|---|---|---|---|---|
| | | | Math | GSM8K | Svamp | NumGLUE | DM | Simuleq | Aqua | Sat | MMLU-M |
| Base | 7B | 2.9 | 3.4 | 3.2 | 14.5 | 16.7 | 7.5 | 3.3 | 6.7 | 8.6 | 5.5 |
| E2E LoRA | 7B | **0.56** | **16.3** | 30.9 | 59.0 | **37.9** | **37.1** | **9.1** | **25.6** | **30.5** | **34.3** |
| Local LoRA | 7B | 0.68 | 3.6 | 7.9 | 30.6 | 21.0 | 15.9 | 1.9 | 13.0 | 13.6 | 18.4 |
| Local LoRA | 13B | - | 11.1 | **31.7** | **62.2** | 34.9 | 26.2 | 5.6 | 24.4 | 22.3 | 27.7 |
| E2E LoRA (only last chunk) | 7B | 0.69 | 2.1 | 1.6 | 3.5 | 5.2 | 3.5 | 0.4 | 5.9 | 13.2 | 8.3 |

## 4 Experiments

**Fine-Tuning Task** We fine-tune the Llama 2 7B and 13B models (Touvron et al., 2023) on MathInstruct (Yue et al., 2023), a state-of-the-art instruction-tuning dataset for general math problem-solving. In the case of the 7B model, fine-tuning on this dataset was shown to outperform the then best open-source model (WizardMath) by 25%. We evaluate open-ended reasoning on Math (Hendrycks et al., 2021), GSM8K (Cobbe et al., 2021), SVAMP (Patel et al., 2021), NumGLUE (Mishra et al., 2022), DeepMind (Davies et al., 2021), and Simuleq (Koncel-Kedziorski et al., 2016). We also evaluate multiple-choice question answering on AQuA-RAT (Ling et al., 2017), SAT-Math (Zhong et al., 2023), and MMLU-Mathematics (Hendrycks et al., 2020) datasets.

**Hyper-Parameters** We select a learning rate of $4 \times 10^{-5}$ using a small grid search (see the Appendix). We use a batch size of $64$, a cosine learning rate scheduler with a warm-up ratio of $0.03$, and train for one epoch. We apply LoRA to all linear layers, except the de-embedding layer, and set the rank $r = 8$. We quantize the pre-trained model weights to 8 bits, and use gradient checkpointing.

### 4.1 Results

Table 1 compares the performance of the pre-trained 7B model, and the 7B model fine-tuned with E2E LoRA and Local LoRA (split into two chunks of 16 blocks each). Despite the simple proxy model, we find that Local LoRA improves performance over the pre-trained base model, although it falls short of E2E LoRA. We also report the performance of the 13B model fine-tuned with Local LoRA (split into two chunks of 20 blocks each), finding that on some tasks Local LoRA can match the performance of E2E LoRA. This demonstrates that, by using Local LoRA, we can fine-tune a larger base model and potentially achieve better downstream performance. For example, the Nvidia RTX 3080 is a common consumer GPU with 10GB memory. If this is the GPU the user has access to, they would not be able to perform E2E fine-tuning of the 13B model, under our configuration, but they would be able to fine-tune using Local LoRA.

**Ablations** Figure 2 studies the evaluation loss of the model as we split the 7B model into an increasing number of chunks. The performance decreases as the model is split further, likely because the small proxy model achieves a poorer approximation of the gradient as it approximates more and more of the model. However, dividing the model in more chunks allows fine-tuning of a larger base model, the improved capability of which may overcome this decrease.

Our next two experiments confirm that the gradient signal provided by the proxy model is useful for learning. Figure 3 shows the training loss over time of each chunk of the 7B model split into four chunks, revealing that each chunk successfully learns using the proxy model. Second, the last row of Table 1 shows the downstream performance of an alternative training scheme where the model is split into two chunks, but only the second chunk is fine-tuned and the first chunk is held at the pre-trained weights. This approach performs poorly, demonstrating the importance of updating the first chunk with the loss loss function.

Finally, we record the training times for the 7B model: On an A100 40GB GPU, E2E LoRA took 22.0h, while Local LoRA with two chunks took 27.7h, three chunks 29.8h, and four chunks 32.1h.

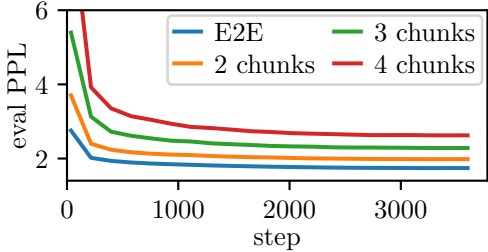
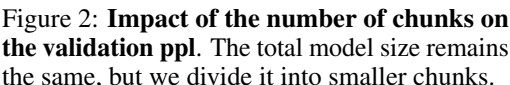
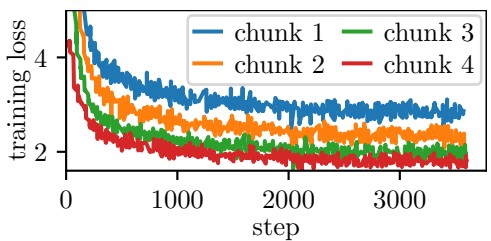

Figure 2: **Impact of the number of chunks on the validation ppl**. The total model size remains the same, but we divide it into smaller chunks.

Figure 3: **Training loss of each chunk of a four-chunk model**. Each chunk successfully learns using the proxy model.

## 5 Related Work

**Parameter-Efficient Fine-Tuning (PEFT)**    Vanilla fine-tuning of entire LLMs requires the same amount of memory as pre-training, rendering it infeasible for many practitioners (Kaddour et al., 2023a). PEFT refers to a class of methods that adapt LLMs by updating only a small subset of model parameters, thereby being more memory-efficient (Houlsby et al., 2019; Ding et al., 2022; Lialin et al., 2023). Popular approaches include prompt-tuning Li & Liang (2021); Lester et al. (2021), adapters (Houlsby et al., 2019; Pfeiffer et al., 2020; Sung et al., 2022), low-rank-parameterized update matrices (Hu et al., 2021; Dettmers et al., 2023), and scaling activations by learned vectors (Liu et al., 2022). Local LoRA is orthogonal to (and compatible with) these approaches since it does not restrict which target parameters to fine-tune (be it full layers, adapters, low-rank update matrices, etc.). This work focuses on extending LoRA since it is arguably the most popular PEFT method. For future work, it would be interesting its framework to other PEFT methods and compare their results.

**Model Parallelism**    Local LoRA is closely related to local learning work on model-parallel distributed training (Jaderberg et al., 2017; Mostafa et al., 2018; Nøkland & Eidnes, 2019; Belilovsky et al., 2020; Laskin et al., 2020). These approaches use a loss function local to each accelerator to provide a gradient signal for the model chunk stored on that accelerator in a similar fashion to how LoFT uses a proxy model to generate a gradient signal for the currently loaded chunk. The motivation is to scale the training across a large number of accelerators while avoiding the communication of gradients between them, which would otherwise slow down the training. In contrast, LoFT aims to make fine-tuning of large models possible with only a small amount of resources by estimating the gradient signal from the unloaded portion of the model.

**Local Learning**    Local objectives appear in the wake-sleep algorithm (Hinton et al., 1995) and more recent algorithms that attempt to implement biologically plausible learning (Löwe et al., 2019; Hinton, 2022). They were also used in Inception networks (Szegedy et al., 2015) to solve the vanishing gradient problem.

## 6 Conclusion and Future Work

The primary limitation of our initial work is the simple choice of proxy network. We plan to explore creating improved proxy networks by performing supervised training of the proxy to match the gradients of the full network, taking inspiration from Jacobian matching in the knowledge distillation literature (Hinton et al., 2015; Czarnecki et al., 2017).

## Acknowledgements

JK and OK acknowledge support from the Engineering and Physical Sciences Research Council with grant number EP/S021566/1. OK was supported by G-Research. PM was partially funded by the European Union's Horizon 2020 research and innovation programme under grant agreement no. 875160, ELIAI (The Edinburgh Laboratory for Integrated Artificial Intelligence) EPSRC (grant no. EP/W002876/1), an industry grant from Cisco, and a donation from Accenture LLP; and is grateful

to NVIDIA for the GPU donations. This work was supported by the Edinburgh International Data Facility (EIDF) and the Data-Driven Innovation Programme at the University of Edinburgh.

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

# A    Grid search

Table 2: **Grid search to select the learning rate for LoRA and Local LoRA.** In both cases we choose $4 \times 10^{-5}$

| Learning rate | Local LoRA | LoRA (E2E) |
|---|---|---|
| $2 \times 10^{-5}$ | 2.089 | 1.785 |
| $3 \times 10^{-5}$ | 2.029 | 1.758 |
| $4 \times 10^{-5}$ | 1.983 | 1.743 |

