# OpenReview forum: "Local LoRA: Memory-Efficient Fine-Tuning of Large Language Models"
_NeurIPS.cc/2023/Workshop/WANT — WANT@NeurIPS 2023 Poster_

### Official Review · Reviewer_5uaA · 2023-10-24
**Local-LoRA is promising but need more delicate design**

**Confidence:** 3

**Review:**

The paper addresses an important practical issue in fine-tuning large language models - the memory constraints that prevent fine-tuning the full model end-to-end on typical hardware. Here are the pros and cons for this paper:

Pros:

- Local LoRA can improve over an unfine-tuned baseline and close the gap with end-to-end fine-tuning, especially when using more chunks to fit a larger base model.
- The approach is general and could likely be applied to other parameter-efficient fine-tuning methods besides LoRA.

Cons:

- Performance decreases as model is split into more chunks, likely due to limitations of the simple proxy model used. Moreover, the method introduces some stale gradient issues since embeddings are computed on older versions of previous chunks.
- Training time increases when using more chunks, reducing the efficiency gains.

---

### Official Review · Reviewer_JwZV · 2023-10-25
**The paper introduces an innovative method for fine-tuning large language models on devices with limited memory. It achieves this by employing model and dataset chunking.**

**Confidence:** 3

**Review:**

[Strengths]

One of the notable strengths of the paper is its proposal of a straightforward yet effective approach for finetuning large models when dealing with limited GPU memory. This approach is supported by the experimental results presented in the paper.

[Weaknesses]

While the paper provides a clear rationale for the choice of the proxy model, it would be beneficial to include a comparison with other alternative options. This would help to understand quality drop compared to the E2E LoRA approach.

[Novelty and significance]

The paper introduces a novel idea of tuning the model chunk by chunk to overcome the limitation of uploading the entire model to the GPU. Furthermore, the paper addresses the challenge of handling large datasets by proposing a chunking solution.

[Technical quality and empirical evaluation]

The paper demonstrates good technical quality with well-written content and presents experimental results that generally support the main claims. However, the comparison is limited to the E2E LoRA approach, and it would be beneficial to include additional approaches for a more comprehensive analysis. Additionally, more detailed explanations and analysis regarding the performance discrepancy between the Local LoRA 7B and Base models on the Simuleq dataset would provide valuable insights.

---

### Official Review · Reviewer_GywR · 2023-10-25
**A block coordinate training of LoRA adapters for better memory efficiency.**

**Confidence:** 4

**Review:**

Authors suggested to train LoRA adapters in following manner: we split all trainable parameters in chunks from tail to head of Transformer-model, and train these parameter chunks one by one in the same order from tail to head of the model. In this case we can save compute with reused output activation from previous parameter chunk. With combination to other memory saving methods like offloading and/or checkpointing, Local LoRA could have practical interest in fine-tuning edge and mobile device or in IoT.

## Strengths

1. Clear and simple idea which brings ALS-like iterations for optimization objective function over subsets (transformer-blocks) of entire set of parameters.
2. It is complementary with other methods for improve memory efficiency albeit it requires additional efforts.
3. A quite sound experiment setup and ablation study. It answers to questions (a) what best values of performance metrics can be achieved and (b) how training depends on number of chunks.

## Weaknesses

1. Typesetting issues and broken paper high-level structure: missing conclusion section, related works in appendix. It seems that authors were in hurry right before deadline and got bad-structured document.
2. Limited Experiments section: Local LoRA are compared only again pretrained and fine-tuned end-to-end LLaMA 2. It would be great to see experimental comparison with checkpointing and offloading methods. Hyperparameters are picked up not very thouroughtly (only 4 points are tested).
3. Unclear implementation details. It seems that proper implementation should somehow reinitialize optimizer states or offload it to CPU for each chunk. It is quite an important issues in specific experimental setups (e.g. small batch, small sequences, large ranks, optimizer state has the biggest share in used memory).
4. It is not clear limitations of the proposed methods and its practical applicability.

---

### Official Review · Reviewer_gjT3 · 2023-10-25
**Interesting memory-efficient fine-tuning technique with room for improvement**

**Confidence:** 3

**Review:**

**Summary**

The paper introduces Local LoRA, a method to fine-tune arbitrarily large models on fixed hardware, including a single consumer-grade GPU. The key idea is to divide the model into chunks and fine-tune each chunk sequentially, thus decoupling the model size from memory requirements.

**Contributions**

- The paper proposes an intuitive and simple approach to fine-tune a large model chunk-by-chunk which enables to theoretically fine-tune arbitrarily large models on a single GPU.
- It proposes a simple proxy model as a loss function, which computes the gradients for a chunk $M_i$ without having access to the gradients of subsequent chunks.
- The algorithm also divides the training dataset into chunks and tokenizes the current dataset chunk on the fly during the first model chunk's iteration. This strategy reduces memory requirements for the embeddings.
- The authors evaluate their Local LoRA approach by fine-tuning a LLaMA2 7B and 13B parameter model.

**Cons**

- The results from the Local LoRA LLaMA 7B and 13B fine-tuning, especially with a larger chunk count, are not overly encouraging. The Local LoRA 13B fine-tuning even lags behind the E2E LoRA 7B solution in most benchmarks.
- Local LoRA’s performance seems to detoriate fast with increasing the number of model chunks, suggesting the approaches impractability to fine-tune even larger models than the ones evaluated in this work.

**Correctness**

There seems to be a noticeable difference between the MATH and GSM8k benchmark results reported for the base LLaMA 7B model in this paper and those in Touvron et al. (2023). Specifically, the GSM8k scores are 14.6 instead of 3.2, and the Math scores are 2.5 instead of 3.4. Releasing the precise evaluation setup would be beneficial in reproducing these results.

**Clarity**

The paper is very well written. Figure 1 provides a good intuitive understanding of the Local LoRA approach.

**Reproducibility**

The paper provides details for the reproduction of results, mentioning the models, hyperparameters and datasets used, and providing pseudocode for the algorithms. To further improve reproducibility, it would be beneficial if the authors made their code, checkpoints and evaluation setup available to the public.

**Additional Comments:**

- The paper mentions in Section 4.1 that the E2E LoRA fine-tuning couldn't be done on an RTX 3080 GPU. However, the actual training runs were conducted on an A100 GPU, as stated on line 126.
- Also, was the training done with a 40GB or 80GB A100 GPU?

****Small notes:****

- l. 23-25: The sentence “Thus, the entire pre-trained network must be stored in memory.” is repeated two times.
- Appendix B, Table 2: It says that $1 \times 10^{-5}$ was selected as the learning rate, but this learning rate is not showing up in the table.

Overall, the paper offers an interesting technique for fine-tuning arbitrarily large models on fixed hardware, or even just a single consumer GPU. However, the detoriated performance compared to LoRA fine-tuning indicate potential areas for improvement.

---

### Meta-Review · Area_Chair_5ABq · 2023-10-27

**Recommendation:** Accept (Poster)
**Confidence:** 4

**Metareview:**

**Strengths:**
* Most reviewers found the contributions to be novel.
* Paper is well-written and easy to follow.
* Solves a relevant problem using a straightforward and easy to implement approach.
* Fairly strong evaluation with good results and ablation studies. However, some reviewers pointed out that the evaluation should be expanded to cover more models and techniques such as checkpointing/offloading.

**Weaknesses:**
* Mixed results: Local LoRA finetuning of the 13B model underperforms E2E LoRA 7B on some benchmarks.
* Scalability: not clear how approach scales to larger models and/or larger number of chunks. Training time also appears to increase with more number of chunks.
* Mismatched results for MATH and GSM8k compared to Touvron et al.
* Issues with formatting and paper structure.

The overall sentiment for the paper based on the submitted reviews appears to be largely positive. I recommend acceptance (poster).

---

### Decision · Program_Chairs · 2023-10-28

**Decision:**

Accept (Poster)

**Comment:**

We thank the authors for their time and contribution to WANT and we are pleased to share that after the reviewing process the paper has been accepted. Congratulations! We encourage the authors to consider reviewers' feedback for the improvement of the camera-ready version. We hope to see you in person at the workshop and brainstorm on efficient training research together!